# Reciprocal Interactions between Fibroblast and Pancreatic Neuroendocrine Tumor Cells: Putative Impact of the Tumor Microenvironment

**DOI:** 10.3390/cancers14143481

**Published:** 2022-07-18

**Authors:** Thomas Cuny, Peter M. van Koetsveld, Grégoire Mondielli, Fadime Dogan, Wouter W. de Herder, Anne Barlier, Leo J. Hofland

**Affiliations:** 1Department of Internal Medicine, Division of Endocrinology, Erasmus Medical Center, 3015 GD Rotterdam, The Netherlands; thomas.cuny@ap-hm.fr (T.C.); p.vankoetsveld@erasmusmc.nl (P.M.v.K.); f.dogan@erasmusmc.nl (F.D.); w.w.deherder@erasmusmc.nl (W.W.d.H.); 2Department of Endocrinology, Hôpital de la Conception, Centre de Référence des Maladies Rares Hypophysaires HYPO, Hôpitaux Universitaires de Marseille, 13005 Marseille, France; anne.barlier@univ-amu.fr; 3DiPNET Team, U1251, INSERM, Marseille Medical Genetics, Aix-Marseille Université, CEDEX 05, 13385 Marseille, France; gregoire.mondielli@univ-amu.fr; 4Laboratory of Molecular Biology, Hôpital de la Conception, Hôpitaux Universitaires de Marseille, 13005 Marseille, France

**Keywords:** neuroendocrine tumor, microenvironment, fibroblasts, cell lines

## Abstract

**Simple Summary:**

Treatment of pancreatic neuroendocrine neoplasms (PNEN) is challenging since a subset of patients will have a metastatic disease at diagnosis and/or will experience resistance to the treatment. Whether the tumor microenvironment (TME) may represent a therapeutic target of interest in these tumors, remains to be elucidated. Our aim was to investigate the role played by stromal fibroblasts (namely, HPF and HFL-1) over the growth of human PNEN cell lines BON-1 and QGP-1, and vice versa. We confirmed a reciprocal stimulatory effect of HPF and HFL-1 over the growth of BON-1 and QGP-1, and the other way around. Likewise, the number of BON-1/QGP-1 colonies and the migration potency of both cell lines, significantly increased in presence of fibroblast-conditioned medium. Finally, the mTOR inhibitor, everolimus, mitigated the fibroblast-induced stimulatory effect over BON-1/QGP-1 cell lines, suggesting that fibroblasts, as an actor of the TME of PNEN, is a promising therapeutic target.

**Abstract:**

Introduction: Pancreatic neuroendocrine neoplasms (PNENs) present with a fibrotic stroma that constitutes the tumor microenvironment (TME). The role played by stromal fibroblasts in the growth of PNENs and their sensitivity to the mTOR inhibitor RAD001 has not yet been established. Methods: We investigated reciprocal interactions between (1) human PNEN cell lines (BON-1/QGP-1) or primary cultures of human ileal neuroendocrine neoplasm (iNEN) or PNEN and (2) human fibroblast cell lines (HPF/HFL-1). Proliferation was assessed in transwell (tw) co-culture or in the presence of serum-free conditioned media (cm), with and without RAD001. Colony formation and migration of BON-1/QGP-1 were evaluated upon incubation with HPFcm. Results: Proliferation of BON-1 and QGP-1 increased in the presence of HFL-1cm, HPFcm, HFL-1tw and HPFtw (BON-1: +46–70% and QGP-1: +42–55%, *p* < 0.001 vs. controls) and HPFcm significantly increased the number of BON-1 or QGP-1 colonies (*p* < 0.05). This stimulatory effect was reversed in the presence of RAD001. Likewise, proliferation of human iNEN and PNEN primary cultures increased in the presence of HFL-1 or HPF. Reciprocally, BON-1cm and BONtw stimulated the proliferation of HPF (+90 ± 61% and +55 ± 47%, respectively, *p* < 0.001 vs. controls), an effect less pronounced with QGP-1cm or QGPtw (+19 to +27%, *p* < 0.05 vs. controls). Finally, a higher migration potential for BON-1 and QGP-1 was found in the presence of HPFcm (*p* < 0.001 vs. controls). Conclusions: Fibroblasts in the TME of PNENs represent a target of interest, the stimulatory effect of which over PNENs is mitigated by the mTOR inhibitor everolimus.

## 1. Introduction 

Human gastroenteropancreatic neuroendocrine neoplasms (GEPNENs) represent a heterogeneous group of tumors, as they are derived from neuroendocrine cells that are widely distributed throughout the body. They currently account for approximately 0.5% of all human cancers, with an increasing incidence over the past 30 years that has reached around 3–5/100,000 individuals per year [1,2,3,4]. Patients with pancreatic neuroendocrine neoplasms (PNENs) represent around 10% of all patients with neuroendocrine neoplasms seen in the clinic [3]. The current treatment for GEPNENs consists of a multimodal approach, and surgery remains, if feasible, the first therapeutic option that can completely cure the disease. Besides surgery, locoregional treatment and chemo- and/or radiotherapy, as well as targeted therapies, represent alternative options that are generally discussed on a case-by-case approach [5,6,7]. In spite of the substantial progress that has been made in the early diagnosis and the therapeutic management of GEPNENs, about half of patients develop metastasis, mainly in the liver, either at the initial diagnosis or during the follow-up [3]. Currently, the mTOR inhibitor everolimus (or RAD001) is approved for the treatment of neuroendocrine neoplasms from digestive, pancreatic and lung origins [8,9]. The rationale of its use in PNENs relies on the existence of overactivation of the mTOR signaling pathway in neuroendocrine tumoral cells [10,11]. However, the pharmacological effects of RAD001 in the surrounding cells of the primitive tumor, known as the tumor microenvironment (TME), remain elusive. The influences exerted by the TME on the sensitivity-to-treatment of the tumor are still poorly understood and could represent an explanation for the limited in vivo efficacy of RAD001 in inhibiting tumor progression or inducing tumor stabilization, which contrasts strikingly with its efficacy in vitro [10,12]. 

The TME is a complex structure comprising multiple cell types, a supportive matrix and soluble factors, which promote tumor growth and influence its behavior [13,14]. Among them, cancer-associated fibroblasts (CAFs) represent a key cellular component in the TME. Their role in the initiation of cancer invasion, although not completely understood, is likely to be the consequence of oncogenic signals that the CAF can provide to the tumoral cells in a paracrine fashion [15]. Today, the specific role played by the CAFs within the TME of GEPNENs remains to be elucidated. The frequent occurrence of a remarkable fibroblastic reaction in GEPNENs [16,17], *a fortiori* in the case of a small intestine initiation site, suggests that an important infiltrate of fibroblasts or fibroblast-like cells is likely to occur in the TME of GEPNENs. 

The purpose of the present study was to determine whether human fibroblasts can stimulate, in vitro, the proliferation and/or the migratory capacity of human PNEN cells. 

We assessed interactions between human PNEN cell lines (BON-1 and QGP-1) and primary cultures of two human GEPNENs, on the one hand, and two human fibroblast cell lines, HFL-1 and HPF, on the other hand. Two different approaches were used to evaluate the cellular interactions: (1) by co-incubation of either BON-1 or QGP-1 with HFL-1 or HPF cells using permeable supports in a Boyden chamber to obtain a co-culture system without a “physical” contact between cell lines, and (2) by culturing BON-1 and QGP-1 cell lines with fibroblast-conditioned medium and, reciprocally, fibroblasts with PNEN-conditioned medium. Finally, we assessed the sensitivity of both BON-1 and QGP-1 cells when treated with the mTOR inhibitor everolimus (RAD001) in the presence of the fibroblast-conditioned medium or fibroblasts with PNEN-conditioned medium. 

## 2. Material and Methods 

### 2.1. Cell Lines 

BON-1 is a permanent cell line derived from a lymph node metastasis of a human carcinoid tumor of the pancreas [18]. It was a kind gift from Dr Townsend (University of Texas Medical Branch, Galveston, TX, USA). The QGP-1 cell line, which is derived from a pancreatic islet cell carcinoma, was purchased from the Japanese Collection of Research Bioresources (JCRB) Cell Bank [19]. HFL-1 cell is a fibroblast cell strain derived from the lungs of a male human fetus of 16 weeks’ gestation [20], which was purchased from the Global Bioresource Research Center ATCC (ATCC^®^ CCL-153^TM^, Manassas, VA, USA). The commercially available human fetal primary pancreatic fibroblast cell line HPF was purchased from Vitro Biopharma (Vitro Biopharma, Golden, CO, USA). 

All cell lines (BON-1, QGP-1, HFL-1 and HPF) were cultured in 1:1 mixture of DMEM and Ham’s F-12K (Kaighn’s) medium (hereafter called DMEM/F12), supplemented with 10% (*v*/*v*) fetal calf serum (FCS), penicillin (1 × 10^5^ units/L), fungizone (0.5 mg/L) and L-glutamine (2 mmol/L). HPFs were conditioned to grow in DMEM/F12 only after passage 2 (P2). For passage 1, this cell line was cultured in a low-serum mesenchymal stem cell culture medium, MSC-Gro™, purchased from Vitro Biopharma (SC00B1-1) and supplemented with penicillin (1 × 10^5^ units/L), fungizone (0.5 mg/L) and L-glutamine (2 mmol/L), according to the manufacturer. By adding a small amount (20% *v*/*v*) of DMEM/F12 medium, we made sure that the HPFs were used to growing in DMEM/F12. Ultimately, we switched entirely to complete DMEM/F12 at day 7 after trypsinization. In the experimental protocol, NEN cell lines were used before passage 5 and fibroblasts before passage 3. All cells were placed in a humidified incubator at 5% CO_2_ and 37 °C. The media and supplements were obtained either from Life Technologies (Invitrogen) or from Vitro Biopharma. 

### 2.2. Collection and Processing of Serum-Free Conditioned Medium

Serum-free conditioned media were collected from BON-1, QGP-1, HFL-1 and HPF cell lines, previously grown in 75 cm^2^ cell culture flasks by Corning^®^ (Amsterdam, The Netherlands). 

***Fibroblast-conditioned medium*.** For both HFL-1 and HPF, 5 × 10^5^ cells were seeded in 10 mL of serum-enriched DMEM/F12 and placed in the incubator with media changes every 72 h. At day 7, cells were carefully washed using tempered phosphate-buffered saline (PBS, pH 7.4) and 10 mL of serum-free DMEM/F12 medium containing 0.2% bovine serum albumin (BSA) was added to the flask. The serum-free medium conditioned by either HFL-1 (HFL-1cm) or HPF (HPFcm) was collected after 72 h of incubation. 

***PNEN cell line-conditioned medium*.** BON-1 and QGP-1 were seeded in 10 mL of DMEM/F12 at a density of 2 × 10^5^ and 3 × 10^5^ cells, respectively. Once the growth reached 50% confluency within the flask, the serum-enriched medium was removed and replaced by 10 mL of DMEM/F12 + 0.2% BSA. The corresponding conditioned media (BON-1cm and QGP-1cm, respectively) were collected 72 h later. In every condition, the collected medium was centrifuged to eliminate cell residues before being stored at −20 °C. 

### 2.3. Cell Culturing

Cell lines were passaged weekly by trypsinization with trypsin/EDTA (0.05%/0.53 mM) and resuspended in the culture media described above. Trypan blue staining was used to assess cell viability, which always exceeded 95%. Before plating, cells were counted microscopically in a standard hemocytometer. 

***Conditioned-media experiments.*** For conditioned-media experiments, BON-1 cells (5000/well) and QGP-1 cells (7000/well) were seeded in 24-well multiwell culture plates (Corning) with 1 mL/well of DMEM/F12. At day 4, the cells were incubated either with 1 mL DMEM/F12 + BSA 0.2% alone (control conditions) or with HFL-1cm or HPFcm and DMEM/F12 + 0.2% BSA in a 1:1 ratio (1 mL total volume). Media were changed after 72 h, and DNA measurements were made after a total of seven days of incubation. In similar conditions, BON-1 (5000/well) and QGP-1 (7000/well) were incubated either with 1 mL DMEM/F12 + BSA 0.2% alone (control conditions) or with HPFcm and DMEM/F12 + 0.2% BSA in a 1:1 ratio (1 mL total volume) at day 4, and determination of the daily proliferation rate was performed for a total of seven days.

The effects of BON-1cm and QGP-1cm were evaluated in HPF cells, which represented the most representative fibroblast cell line of the pancreas. The latter were seeded at a density of 15,000 cells/well in 24-well multiwell culture plates (Corning). After 24 h, the medium was removed and replaced either with 1 mL DMEM/F12 + 0.2% BSA alone (control conditions) or with BON-1cm or QGP-1cm and DMEM/F12 + 0.2% BSA in a 1:1 ratio (1 mL total volume). Proliferation of the HPF cells was evaluated after 72 h of incubation with the Cell Titer Glo ^®^ 2.0 assay (Promega, Charbonnières, France).

***Co-culture experiments***. Before co-culture, both the PNEN and fibroblast cell lines were washed prior to incubation and cultured in serum-enriched medium (DMEM/F12) for 72 h. BON-1 cells (5000/well) and QGP-1 cells (7000/well) were then cultured in 24-well multiwell culture plates in the “lower” compartment of a Boyden chamber (Corning^®^ Transwell^®^, ThermoFisher, Waltham, MA, USA). After 72 h, BON-1/QGP-1 cells were co-incubated in DMEM/F12 + 0.2% BSA in the presence of HFL-1 cells (HFL-1tw) or HPF cells (HPFtw) placed in the “upper” compartment at a density of 25,000/insert (Figure 1). QGP-1 and BON-1 were harvested after 7 days of co-incubation and DNA measurements were performed. 

***Primary tumor cell culture.*** Tumor fragments of two human GEPNENs (one originated from the ileum and the other from the pancreas, both grade 1 according to the WHO classification [21]) obtained after surgery were dissociated mechanically and enzymatically with collagenase, as previously described [22]. The tumoral cells were seeded at a density of 5 × 10^4^ cells on 24-well plates coated with extracellular matrix (ECM) (from bovine endothelial corneal cells, as previously described [23] and required for cell adhesion [24]). Cells were maintained in culture with L-valine-depleted DMEM (L-valine was replaced with D-valine to block fibroblast proliferation) and supplemented with 10% FCS, penicillin (100 U/mL), streptomycin (100 mg/mL) and glutamine (100 U/mL) at 37 °C in a water-saturated atmosphere containing 5% CO_2_ for 24 h. After this period, the GEPNEN cells were incubated for 72 h either with 1 mL DMEM/F12 + 0.2% BSA alone (control conditions) or with HFL-1cm or HPFcm and 0.2% BSA in a 1:1 ratio (1 mL final volume). Similarly, GEPNEN cells were co-cultured with either HFL-1 or HPF (25,000/insert each) placed in the transwell. Due to their slow growth, we used the cellular viability instead of assessing the proliferation rate. Cellular viability was assessed with the Cell Titer Glo Assay after 72 h for both experiments.

### 2.4. Pharmacological Compounds

We tested the effect of the mTOR inhibitor everolimus (known as RAD001; LC Laboratories, Woburn, MA, USA) on the proliferation of both BON-1 and QGP-1. RAD001 was dissolved in 100% dimethylsulfoxide (DMSO) to a 1 mM concentration and stored at −20 °C before dilution to intermediate concentrations in 40% DMSO and use at 100×. In all the experiments, controls were treated with an equivalent 0.4% vehicle DMSO concentration equivalent to the 0.4% final DMSO concentration in the treatment dilution. For BON-1 and QGP-1 cells, we used the IC_50_ concentrations of RAD001 of 1 nM and 5 nM, respectively, as previously described [25]. HPF cells were also treated with RAD001 at the concentration of 10 nM. The protocol followed the same steps as the ones described in the conditioned media and co-culture experiments. A dose curve response to RAD001 (0.001 to 100 nM) was performed for both cell lines after a total of seven days of incubation with DMEM/F12 + BSA 0.2% (control conditions) or HPFcm and DMEM/F12 + 0.2% BSA in a 1:1 ratio (1 mL total volume). 

### 2.5. Assessment of Cellular Proliferation and Viability

The method was based on a DNA-dependent fluorescence enhancement of a fluorochrome. In short, DNA concentration per well, as a measure of the cell number, was measured using the fluorescent dye Hoechst 33258. Cells were extracted in ammonia (1 M) with 0.2% *v*/*v* TritonX-100. A sonification step was performed for 5 s at amplitude 15 (Soniprep 150; MSE) and a 2 mL assay buffer (100 mmol/L NaCl, 10 mmol/L EDTA, 10 mmol/L Tris; pH 7.0) was added. The solution was then centrifuged at 2000× *g* for 5 min and 20 μL aliquots of the supernatant were mixed with 200 μL Hoechst dye H33258 at 100 μg/L. Excitation and emission wavelengths were set at 350 and 455 nM, respectively, and fluorescence compared to a standard curve of calf thymus DNA (type II, no. D-3636, Sigma-Aldrich, St Louis. MO, USA). Each condition of each experiment was tested in quadruplicate. All experiments were carried out at least two times and gave comparable results.

Total DNA content was used to assess the proliferation of BON-1 and QGP-1 but was not sensitive enough to be used with fibroblasts (HFL-1 and HPF), the proliferation rate of which was very low. Therefore, in the latter, we used a luminescent cell viability assay (Cell Titer-Glo) for measurement.

### 2.6. Western Blot Analysis

The impact of RAD001 and HPFcm on signal transduction pathways was further analyzed using Western blot analysis. For this purpose, 5 × 10^5^ BON-1 or QGP-1 cells were seeded into 6-well plates. After 90 min, cells were washed two times in DMEM/F12 + 0.2% BSA and treated with DMEM/F12 + BSA 0.2% (control condition) or with HPFcm and DMEM/F12 + 0.2% BSA in a 1:1 ratio (1 mL total volume). In both conditions, cells were either treated with RAD001 or not, at the concentration of 1 nM for BON-1 cells and 5 nM for QGP-1 cells. Cells were washed with ice-cold PBS and scrapped in 100 µL of 25 mM HEPES-NaOH, pH 7.4, 150 mM NaCl, 1% NP40 protease inhibitors (2 mM sodium orthovanadate, 1 mM sodium fluoride, 10 mM beta-glycerophosphate, 10 µg/mL aprotinin and leupeptin). Cells were bend-over agitated for 15 min at 4 °C, then centrifuged for 20 min at 10,000× g at 4 °C. Protein concentration was determined with the Dc Protein assay (Bio-Rad) using an Enspire Multimode plate reader (PerkinElmer, Villebon sur Yvette, France), according to the manufacturer protocol. Proteins were diluted in Laemmli buffer to 1–5 µg/µL. After extraction, proteins were separated using 12% or 4–15% SDS-PAGE. Samples (40 µg/lane) were heated for 5 min at 100 °C under reducing and denaturating conditions with 20 mM DTT and 2.5% (*w*/*v*) SDS. Proteins were transferred onto a 0.45 um PVDF Immobilon-P membrane (Millipore, Saint-Quentin-en-Yvelines, France) using a Pierce Power Station (ThermoFisher Scientific), according to the manufacturer’s protocol. Membranes were blocked with TBS 0.005% Tween 20 (TBST) and 5% BSA (1 h at room temperature), then incubated with the following primary antibodies (2 h at room temperature): Rabbit Polyclonal Anti-Phospho (Thr202/Tyr204) Erk1/2 (Cell Signaling Technology #9101), Rabbit Polyclonal Anti-Erk1/2 (Santa Cruz #sc-94), Rabbit Monoclonal Anti-Phospho (Ser235/236) S6 Ribosomal Protein (Cell Signaling Technology #4857), Mouse Monoclonal Anti-S6 Ribosomal Protein (Cell Signaling Technology #2317), Akt Total (Cell Signaling Technology #9272), Phospho Ser473 Akt (Cell Signaling Technology #9271), Mouse Monoclonal, Rabbit Polyclonal Anti-4E-BP1 (Cell Signaling Technology #9644), Rabbit Polyclonal Anti-Phospho Ser65 4E-BP1 (Cell Signaling Technology #9451), Rabbit Polyclonal Anti-SRC (Cell Signaling Technology #2108), Rabbit Polyclonal Anti-Phospho Tyr416 SRC (Cell Signaling Technology #2101) and Anti-GAPDH (Merck Millipore #MAB374). Blots were developed with the SuperSignal West Pico or Femto PLUS Chemiluminescent Substrate (ThermoFisher Scientific). Chemiluminescent signals were detected using a CCD camera and Syngene software (G:BOX, Ozyme, France). Signal vs. background was quantified using NIH Image J software.

### 2.7. Colony-Forming Assay

A total of 250 BON-1 and 500 QGP-1 cells were plated in poly-L-lysine (10 μg/mL)-coated 12-well plates in FCS-containing culture medium (see above). After 24 h, cells were washed twice with culture medium containing 0.2% BSA and 72 h incubations in +0.2% BSA medium (control), with HPF-conditioned medium (HPFcm, see above) and with or without everolimus (1 nM for BON-1 and 5 nM for QGP-1), were initiated in quadruplicate. After 72 h, the media were removed and replaced by FCS-containing medium for 7 days. After 7 days, plates were washed twice with PBS, and cells were fixed for 10 min with 100% ethanol and stained with haematoxylin. Colonies were digitalized using a MultiImage light cabinet (Alpha Innotech, San Leandro, CA, USA) and colony number and size were analyzed using ImageJ software. Plating efficiency (PE) was calculated as the mean number of colonies/number of plated cells for control cultures and for cells incubated with HPFcm or everolimus. The surviving fraction was calculated as the mean number of colonies (number of inoculated cells × PE). The effects on the surviving fraction and colony size represent the cytotoxic and cytostatic effects, respectively. Each condition of an experiment was tested in duplicate. All experiments were carried out at least two times and gave comparable results.

### 2.8. Migration Assay

The impact of HPFcm on the migratory potential of BON-1 and QGP-1 cells was assessed. BON-1 and QGP-1 were seeded (700,000/well and 1,000,000/well, respectively, 3 mL/well) on poly-L-lysine-coated 6-well multiwell plates (Corning) with DMEM/F12. After 4 days, a scratch was manually made with a pipette point and medium was changed to either DMEM/F12 + 0.2% BSA (control conditions) or HPFcm + DMEM/F12 + 0.2% BSA (ratio 1:1). Pictures were taken every 2 h from T0 to T8 h to assess the distance between the two parallel lanes created and analyzed with Image J software. At each time point, four measures/well were performed. Each condition in each experiment was tested in duplicate. To avoid the measurement of cell proliferation, rather than their migration, we arbitrarily decided to perform the measurements for up to 8 h of the experiments and not beyond. Each condition in each experiment was tested in duplicate. All experiments were carried out at least two times and gave comparable results.

### 2.9. Statistical Analysis

For statistical analysis, GraphPad Prism ^®^ version 9 (Graph-Pad Software, San Diego, CA, USA) was used. Comparative statistical evaluations between groups were accomplished with unpaired Student’s *t*-tests or one-way ANOVA, followed by Tukey’s tests for multiple post hoc comparisons. Outliers were excluded using Grubbs’ test with the GraphPad Quick-Calcs outlier calculator. Data are reported as means ± SEM. In all analyses, a two-sided *p* value <0.05 (* *p* < 0.05, ** *p* < 0.01, *** *p* < 0.001, **** *p* < 0.0001) was considered statistically significant.

## 3. Results 

### 3.1. Impact of Fibroblasts on the Proliferation of the BON-1 Cell Line

In conditioned medium experiments, proliferation of the BON-1 cell line significantly increased when incubated with HFL-1cm compared to control (169.7 ± 24.7% vs. 100 ± 19.9%, *p* < 0.0001) (Figure 2A). A comparable result was observed when BON-1 cells were incubated with the HPFcm as compared to control (146.2 ± 18.7% vs. 100 ± 9.9%, *p* < 0.0001) (Figure 2B,E,F). In transwell experiments, the proliferation of BON-1 cells significantly increased in the presence of either HFL-1 (149.6 ± 36.5% vs. 100 ± 14.3%, *p* < 0.0001) or HPF (147 ± 25.6% vs. 100 ± 13.6%, *p* < 0.001) compared to controls (Figure 2C,D). 

Over a period of 7 days, the viability of BON-1 cells was significantly higher in the presence of HPFcm as compared to control conditions (*p* < 0.0001 from day 5 to day 7, Figure 2G). 

### 3.2. Impact of Fibroblasts on the Proliferation of the QGP-1 Cell Line

Similarly, the proliferation of QGP-1 cells significantly increased when incubated with either HFL-1cm (Figure 3A) or HPFcm (Figure 3B,E,F) compared to control (154.6 ± 32.4% vs. 100 ± 10.8% and 142.3 ± 15.4% vs. 100 ± 8.7%, respectively, *p* < 0.0001). In transwell experiments, the proliferation of QGP-1 cells was also significantly stimulated by the presence of either HFL-1 or HPF cells compared to control (144.9 ± 12.1% vs. 100 ± 8% and 158 ± 19.4% vs. 100 ± 15.9%, respectively, *p* < 0.0001 and *p* < 0.01) (Figure 3C,D). 

Over a period of 7 days, we observed a statistically significant higher cellular viability of QGP-1 in the presence of HPFcm as compared to control conditions from day 4 to day 7 (*p* < 0.01, Figure 3G). 

### 3.3. Impact of Fibroblasts on the Colony Formation of BON-1/QGP-1

We assessed the impact of HPFcm on the colony-forming capacity of BON-1 and QGP-1 cells. Interestingly, in both cell lines, the number of colonies significantly increased in the presence of HPFcm as compared to control (100 ± 61% vs. 373 ± 248%, respectively, for BON-1, *p* < 0.01, and 100 ± 18% vs. 195 ± 35% for QGP-1, *p* < 0.05), and this stimulatory effect was reversed by incubating cells with RAD001 (Figure 4A,B). Conversely, we did not observe a similar significant impact from either the HPF-conditioned medium or RAD001 on the size of the colonies (Figure 4A,B).

### 3.4. Impact of Fibroblasts on the Cellular Viability of Primary Cultures of Human Neuroendocrine Neoplasm

The impact of the fibroblast-conditioned media was assessed in primary cultures of two human GEPNENs. In the presence of either HFL-1cm or HPFcm, the cellular viability of the ileal NEN (iNEN) cells significantly increased compared to the control (133.6 ± 27% and 137.8 ± 14.1% for HFL-1cm and HPFcm, respectively, vs. 100 ± 24%, for controls, *p* < 0.01 for both, Figure 5A). A similar trend was observed when iNEN cells were co-cultured with HFL-1 or HPF in transwells (126.6 ± 8.7% and 112.7 ± 13.2% for HFL-1tw (*p* < 0.05) and HPFtw (NS), respectively, vs. 100 ± 4.4% for controls, Figure 5A). In primary PNEN cells, the cellular viability increased in the presence of either HFL-1cm or HPFcm (141.4 ± 16.6% (*p* < 0.001) and 133.8 ± 16.7% (*p* < 0.01), respectively, vs. 100 ± 10.4% for controls, Figure 5B) and in the presence of HFL-1tw or HPFtw (124.9 ±11.7% (NS) and 131 ± 26% (*p* < 0.01), respectively, vs. 100 ± 10.3% for controls, Figure 5B).

### 3.5. Effects of BON-1 and QGP-1 on the Proliferation of HPF

To assess whether a reciprocal interaction may occur between the GEPNEN cell lines and fibroblasts, the proliferation of HPF was assessed either in the presence of BON-1- or QGP-1-conditioned medium or with BON-1 or QGP-1 incubated in a transwell. In conditioned media experiments, the proliferation of HPF significantly increased as compared to control (190.1 ± 61.4% vs. 100 ± 6.96% for BON-1cm, *p* < 0.0001, and 127.2 ± 19.4% vs. 100 ± 4.5% for QGP-1cm, *p* = 0.0018) (Figure 6A,B). Likewise, the proliferation of HPF significantly increased in the presence of BON-1 (155.1 ± 47.4% vs. 100 ± 12.3% for BONtw, *p* < 0.001) and in the presence of QGP-1 (118.8 ± 16.3% vs. 100 ± 15.4%, *p* < 0.05) (Figure 6C,D). As such, the stimulatory effect observed with BON-1 cells was much more potent than that observed with QGP-1 cells.

### 3.6. Impact of RAD001 on the Fibroblast-Induced Proliferation of Neuroendocrine Human Pancreatic Cells and Vice Versa

For drug experiments, we only used conditioned medium experiments, as the results with transwells were roughly superimposable between the two protocols. In BON-1 cells, RAD001 (1 nM) significantly decreased the cellular proliferation in all experiments conducted as compared to the control (Figure 7A: 100 ± 16.2% vs. 58.7 ± 22.4%, *p* < 0.0001, and Figure 7B: 100 ± 9.7% vs. 58.7 ± 22.1%, *p* < 0.0001). Neither the incubation with HFL-1cm or HPFcm resulted in a rescue of the BON-1 cells from the inhibitory effect of RAD001. In the absence of drugs, the increase in BON-1 proliferation was again observed when incubated with either HFL-1cm or HPFcm (156.4 ± 14.8%, *p* < 0.0001 vs. control, and 138.1 ± 7.6%, *p* < 0.001 vs. control, respectively). Interestingly, RAD001 completely abrogated the stimulatory effect of HPFcm over BON-1 cells for concentrations of the drug equal to or exceeding IC50 (Figure 7C).

In the QGP-1 cells, RAD001 (5 nM) resulted in a significant inhibition of the proliferation as compared to control (Figure 7D: 100 ± 5.3% vs. 73.4 ± 6.5%, *p* < 0.01, and Figure 7E: 100 ± 11.4% vs. 78.7 ± 5.4%, *p* < 0.05). Stimulation of the proliferation of QGP-1 also occurred again in the presence of either HFL-1cm or HPFcm as compared to the control (159.4 ± 23.7%, *p* < 0.0001 vs. control for HFL-1cm, and 136.6 ± 13.7%, *p* < 0.0001 for HPFcm, respectively). As for BON-1, the stimulatory effects of HFL-1cm and HPFcm did not persist in the presence of RAD001. Surprisingly, a more pronounced inhibitory effect from RAD001 occurred when the cells were incubated with HFL-1cm (Figure 7E: 73.4 ± 6.5% in QGP-1_RAD001_ without HFL-1cm vs. 43.1 ± 7% of control in QGP-1_RAD001_ with HFL-1cm, *p* < 0.001, respectively). Again, once RAD001 reached its IC50 concentration in QGP-1 cells (5 nM), the stimulatory effect of HPFcm was completely reversed (Figure 7F).

Unlike in PNEN cell lines, RAD001 (10 nM) did not exert a strong inhibitory effect on the proliferation of HPF (Figure 8A,B). As previously observed, in the presence of BON-1cm or QGP-1cm, the proliferation of HPF significantly increased (Figure 8A,B: 214 ± 10.7% vs. 100 ± 3.5% (BONcm) and 121.8 ± 18% vs. 100 ± 14.5% (QGP-1cm), *p* < 0.0001 and *p* < 0.01, respectively), and this stimulatory effect was either partially (with BON-1cm) or completely (with QGP-1cm) reversed in the presence of RAD001 (Figure 8A,B).

### 3.7. Effect of HPFcm on the Signalling Pathways of PNEN Cell Lines

We investigated the effects of HPFcm on the Pi3K/AKT/mTOR and ERK signaling pathways using Western blot analysis in both BON-1 and QGP-1 cells (Figure 9). We did not observe significant differences in the phosphorylated forms of proteins either in the presence of HPFcm or with or without RAD001 (1 nM in BON-1, 5 nM in QGP-1), except for phosphorylated S6, which dramatically decreased in presence of RAD001 in both BON-1 and QGP-1. Likewise, RAD001 treatment was associated with higher levels of pAKT in both cell lines. 

### 3.8. Impact of the HPFcm on the Migration Potency of the NEN Cell Line 

In the presence of HPFcm, BON-1 cell migration increased compared to the control wells (Figure 10A). Interestingly, the migration-promoting effect of the HPFcm occurred from the earliest time point until the end of the experiment. When quantified (the percentage corresponds to the average distance covered by the cells from both sides of the banks) the migration potency was significantly higher at time points T4 (Figure 10A, upper panel) and T8 (Figure 10A, lower panel) in the HPFcm wells compared to control (8.8 ± 2.9% vs. 17.9 ± 1.43% for T4 and 13.0 ± 5.4% vs. 22.5 ± 3.1% for T8, respectively, *p* < 0.0001 for both time points). In QGP-1 cells (Figure 10B), HPFcm also stimulated migration potency at T4 (Figure 10B, upper panel, 3.82 ± 1.7% for control vs. 6.8 ± 1.4% for HPFcm, respectively, *p* < 0.0001) and T8 (Figure 10B, lower panel, 6.76 ± 1.4% for control vs. 12.9 ± 1.9% for HPFcm, respectively, *p* < 0.0001).

## 4. Discussion

In recent years, the concept of the tumor microenvironment (TME) has become a cornerstone in the understanding of tumor behavior, as well as a source for the potential therapeutic targets it contains [13]. In the field of neuroendocrine neoplasms, the TME is known to be highly vascularized [26], which subsequently led to the development and the approval of angiogenesis inhibitors, such as sunitinib malate [27]. In pancreatic neuroendocrine neoplasms (PNENs), it is likely that the TME has other roles as well, in particular because of the plethora of molecular factors (growth factors, cytokines), receptors and endothelial and immune cells that have been described surrounding the primitive tumor niche [28].

In our experimental model, we demonstrate that two human fibroblast cell lines, as actors in the TME, can stimulate the proliferation of two human PNEN cell lines, BON-1 and QGP-1. As such, our study updates the only other study that has functionally assessed interactions that can occur between fibroblasts and PNEN cell lines, this having been published more than 25 years ago [29]. To the best of our knowledge, this study is also the first to demonstrate the stimulatory effect of fibroblasts on proliferation of primary cultures of human GEPNENs. Through our dual, contact-free experimental approaches using conditioned media and a co-culture system, respectively, we provide evidence for the existence of a reciprocal paracrine regulation of this proliferative effect. If we consider that a tumor is made of a primitive niche surrounded by fibroblasts, we can investigate the two aspects of the cellular interaction, from the fibroblasts to the tumor cells (the “centripetal” concept) and from the tumor to the fibroblasts (the “centrifugal” concept). Regarding the later, one study showed that serum-free medium conditioned by the BON-1 cell line stimulated fibroblast proliferation [29]. Our study provides similar data showing a significant increase in fibroblast proliferation under PNEN-conditioned medium. In addition, our data suggest that everolimus, while having a moderate impact on the proliferation of fibroblasts in basal conditions, can significantly suppress the stimulatory effect induced by PNEN-conditioned medium on the proliferation of fibroblasts, suggesting that the mTOR signaling pathway is involved in this stimulatory effect. Previously, Beauchamp et al. showed that TGF-ß, released by BON-1, is a strong activator of fibroblast proliferation [29]. TGF-ß is a cytokine whose biological effects are multiple. In particular, TGF-ß activation led to mTOR activation for collagen synthesis or induction of epithelial mesenchymal transition, two mechanisms that likely occur in the TME of PNENs [30,31,32]. However, the effects of everolimus on human pancreatic fibroblasts have barely been studied until now. Overactivation of Pi3K/AKT/mTOR has been observed in cancer-associated fibroblasts (CAFs) isolated from the exocrine counterpart of pancreatic tumor; namely, human pancreatic ductal carcinoma (PDC) [33]. In animal models of PNENs, treatment with mTOR inhibitor led to modifications to the matrix consistency of the TME and ultimately blocked the dissemination of the tumor and the onset of metastasis [34]. Whether this effect results from modifications in the secretory phenotypes of the fibroblasts that occur under mTOR inhibition is unknown. Consistent with these findings, Duluc et al. showed that inhibition of mTOR in CAFs, isolated from PDC, did not affect their viability but resulted in an antifibrotic effect in the tumor stroma with a decrease in the local release of interleukine-6 (IL-6) [33]. Interestingly, this effect was followed by an improvement in the delivery of chemotherapy to the tumor initiation niche [33]. Whether a similar antifibrotic effect from everolimus exists in the TME of PNENs is still unknown, and the combination of everolimus with chemotherapy and/or antiangiogenic drugs is not currently approved for patients with PNENs. This original concept could pave the way for future research to better delineate which signaling pathways are involved in the proliferation of cancer-associated fibroblasts in PNEN-conditioned medium. Although interesting, these investigations did not constitute the primary goal of our work. 

Our aim was rather to explore whether, and with regard to which aspects, fibroblasts could influence the behavior of pancreatic neuroendocrine cells. All the experiments conducted in our study confirmed that, in the presence of fibroblast-conditioned medium, the proliferation of BON-1 and QGP-1 significantly increases. Likewise, the number of colonies formed by the two cell lines increases as well, although not the colonies’ size. We hypothesize that, in the colony-formation experiments, we are looking at a different population of cells that have colony-forming capacity as compared to experiments in which all cells are grown as a monolayer. These colony-forming cells can survive better under serum-deprived conditions in the presence of fibroblast-conditioned medium, but this does not result in an increase in their proliferation, as illustrated by the absence of any impact on the colony size. Therefore, colony-formation experiments provide data on a different context compared to regular monolayer experiments. The effect of everolimus in these experimental conditions is also interesting since the drug also did not impact the size of the colonies. Which factors are released in high amounts and responsible for a significant biological effect is challenging and hard to establish. We used the whole medium conditioned by the fibroblasts, likely containing hundreds of factors/cytokines synthetized by the cells. However, by doing so, we can also speculate that these experimental conditions better reflect the fibroblast secretions as they are supposed to occur in vivo. In the human neuroendocrine midgut cell line KRJ-I, which was cultured in the upper compartment of a Boyden chamber, Svedja et al. showed that HEK293 fibroblasts placed in the lower compartment synthetized high amounts of profibrotic factors (connective tissue growth factor (CTGF), transforming growth factor beta 1 (TGF-β) and fibroblast growth factor 2 (FGF 2)) [35]. Other studies support the fact that stromal cells in the GEPNENs release high amounts of angiogenic growth factors locally, including vascular endothelial growth factor (VEGF) [36,37], *platelet-derived growth factor* (PDGF) [38] and, again, FGF2 [39].

Our results also stress the fact that the mTOR inhibitor everolimus completely abrogated the stimulatory effect of fibroblast-conditioned medium on the proliferation of human PNEN cell lines. Everolimus has an antiproliferative effect in vitro on PNEN cell lines [40], as well as in primary cultures of human PNEN [22], and it is currently approved for the treatment of well-differentiated PNENs [9]. Besides proliferation, everolimus significantly decreases the number of colonies, data which are in line with the recent work published by Vitali et al. [41], who showed that the number of QGP-1 colonies decreased when treated with everolimus. One limitation of our work is that we have not yet used spheroids to investigate the effect of conditioned medium and/or drugs, such as everolimus. Spheroids represent a preclinical model of reference that reproduces, albeit imperfectly, the molecular and structural organization of in vivo tumors, as compared to monolayer 2D cells [42,43], and can also show significant differences in therapy response when compared to classical 2D culture [44]. Recently, organoid (or tumoroid) models, obtained through an in vitro three-dimensional (3-D) human primary PNEN culture system, have been developed to better characterize the behavior of PNENs isolated from operated patients [45]. Interestingly, this new preclinical model was successfully used for testing sensitivity to different drugs, including everolimus, and could represent an attractive approach for studying the tumor–fibroblast interactions. 

Sensitivity to everolimus is another challenging aspect of the care of patients with PNEN, especially because some of them will eventually develop an acquired state of resistance [46]. In our study, we assessed the potential impact of fibroblast-conditioned medium on the sensitivity to everolimus of PNEN cell lines. Interestingly, our results suggested that everolimus exerted effective inhibitory action on the stimulatory effect of the fibroblast-conditioned medium. These results underline several points for discussion. First, the fibroblast-conditioned medium does not play a role in the acquired resistance to everolimus, a statement which is further underlined by experiments demonstrating that resistance to everolimus, as obtained in vitro, is rather the consequence of chronic exposure to the drug with selection of a resistant clonal tumoral cell [25,47,48]. Moreover, this secondary resistance is likely constituted by the activation of alternative signaling pathways, as has been shown for MYC [49] and even SRC [50,51]. Second, the results of our functional studies, including those using Western blot analysis, suggest that the Pi3K/AKT/mTOR signaling pathway is not overactivated in the presence of fibroblast-conditioned medium. However, crosstalk between signaling pathways in the cells at the basal state or exposed to mTOR inhibitor likely occur [48,50], meaning that no strong conclusion can be established by studying only one signaling pathway in these cells.

Finally, our study shows that the medium conditioned by the fibroblasts can also promote the migration of PNEN cells. The capacity that the tumor cells have to migrate is a complex phenomenon and involves the interaction that the host tumor cells share with the surrounding cells of the TME [52]. Experiments conducted in mice models of PNEN (i.e., the RIP-Tag2 mouse), showed that components of the stroma, such as metalloproteinase 9 (MMP-9) [53] and heparanase [54,55], are key factors for the dissemination of the tumoral cells. 

## 5. Conclusions 

Even though questions remain unanswered and further experiments are needed, our study demonstrates in vitro the existence of reciprocal interactions between human PNEN cells and fibroblasts, where the Pi3K/AKT/mTOR signaling pathway plays a central role. Fibroblasts represent a cell type that occupies a pivotal place within the TMEs of many other tumor types [15], such as their role in pathogenesis, and the subsequent growth of GEPNENs is likely to occur. They represent, therefore, a promising target to consider for the development of therapeutic strategies in the field of GEPNETs.

## Figures and Tables

**Figure 1 cancers-14-03481-f001:**
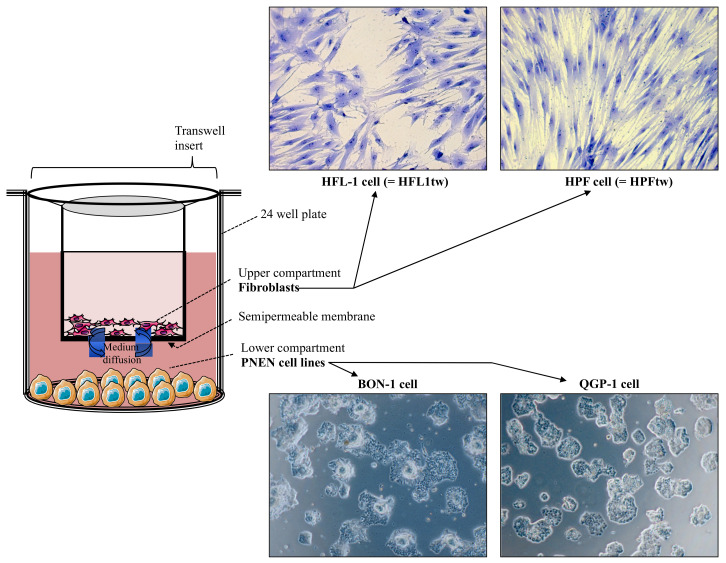
Schematic representation of the co-culture system (Boyden chamber). Human neuroendocrine tumor cells BON-1 and QGP-1 were seeded in the lower compartment, while the HFL-1 and HPF cells were cultured in the upper compartment. The medium, and the produced factors, could diffuse through a semi-permeable membrane.

**Figure 2 cancers-14-03481-f002:**
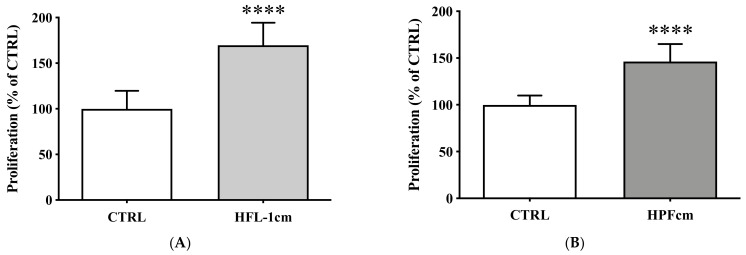
Impact of human fibroblasts on the proliferation of BON-1 cells. (**A**) Increase in the proliferation of BON-1 when incubated with HFL-1cm (**A**) or HPFcm (**B**) compared to the controls. The proliferation of BON-1 cells also increased significantly in the presence of either HFL-1 cells (HFL-1tw (**C**)) or HPF cells (HPFtw (**D**)) in the corresponding transwell experiments. (**E**) Picture taken of BON-1 in control situation. Note the increased density of the cell layer when BON-1 was incubated with the HPFcm (**F**). (**G**) Cellular viability of BON-1 cells in the presence (□) of HPFcm as compared to control (•) over a 7 day period. **** *p* < 0.0001; *** *p* < 0.001.

**Figure 3 cancers-14-03481-f003:**
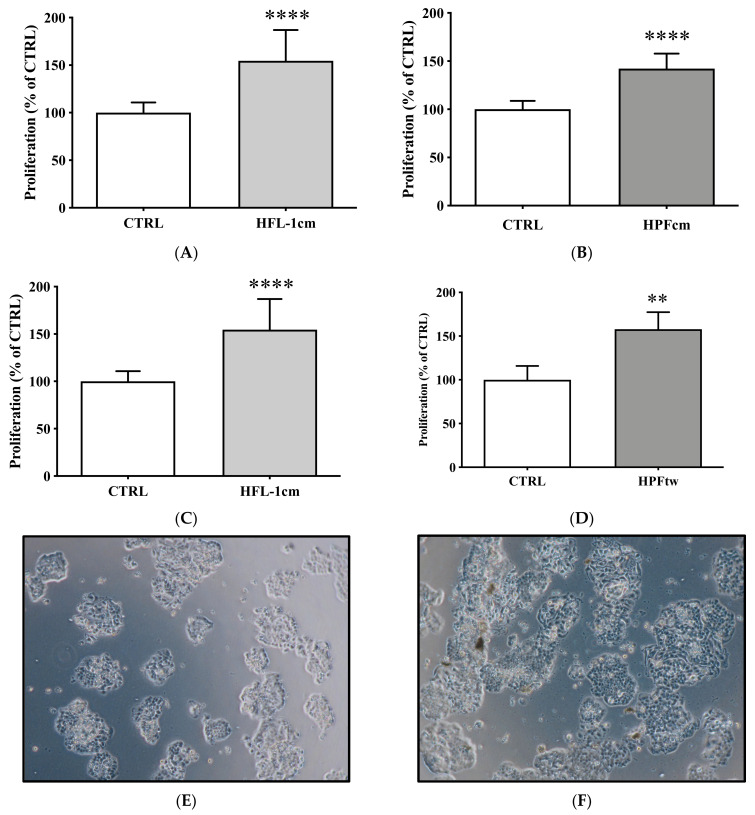
Impact of human fibroblasts on the proliferation of QGP-1 cells. Incubation of QGP-1 with either HFL-1-conditioned medium (**A**) or HPF-conditioned medium (**B**) resulted in a significant increase in proliferation compared to controls. Similar results were observed when QGP-1 was co-cultured with either HFL-1 (**C**) or HPF (**D**) cells in transwell experiments. (**E**) Picture taken of QGP-1 cells in control situation. Note the increased density of the cell layer when QGP-1 was incubated with the HPFcm (**F**). (**G**) Cellular viability of QGP-1 cells in the presence (□) of HPFcm as compared to control (•) over a 7 day period. **** *p* < 0.0001; *** *p* < 0.001; ** *p* < 0.01.

**Figure 4 cancers-14-03481-f004:**
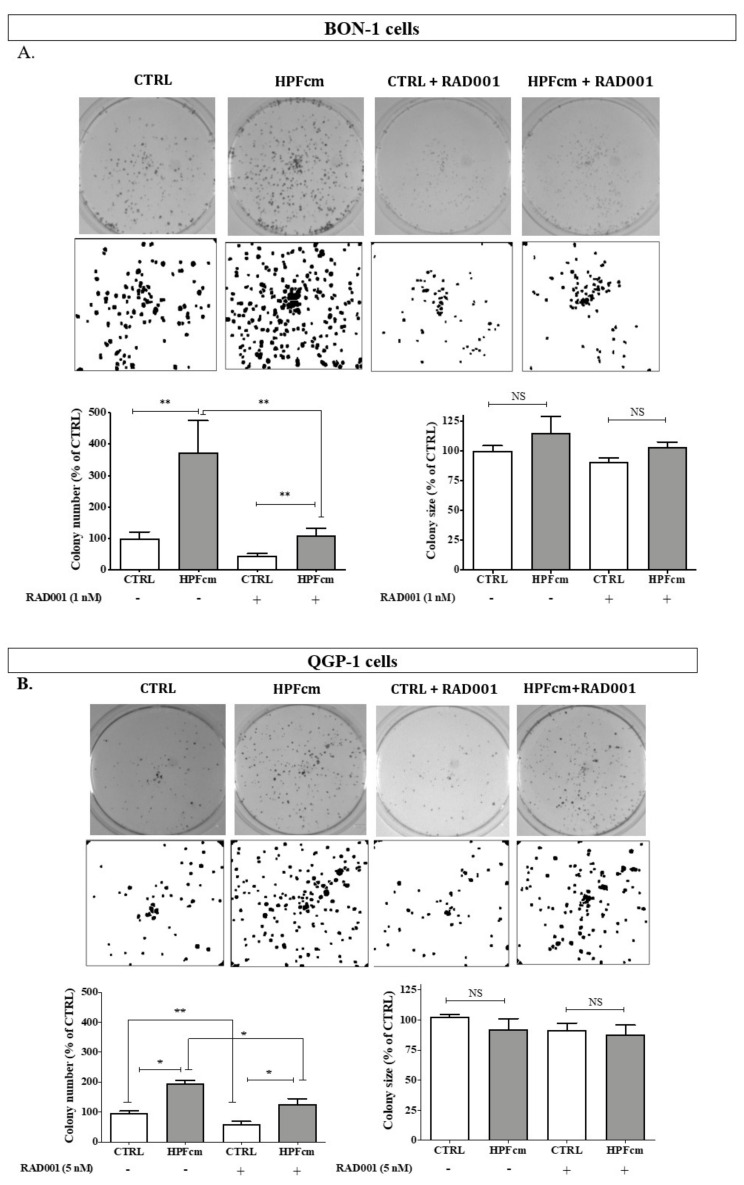
Colony-formation assays of PNEN cells. Incubation with HPFcm resulted in higher numbers of BON-1 colonies, which was reversed by adding RAD001 (1 nM), while no effect from either HPFcm or RAD001 was observed on the size of the colonies ((**A**) and pictures in the upper panel). Similarly, the number of QGP-1 colonies increased in the presence of HPFcm but the effect of RAD001 (5 nM) was less pronounced than in BON-1 ((**B**) and pictures from the lower panel). Again, the size of the colonies was not affected by either RAD001 or HPFcm. ** *p* < 0.01; * *p* < 0.05; NS: not significant.

**Figure 5 cancers-14-03481-f005:**
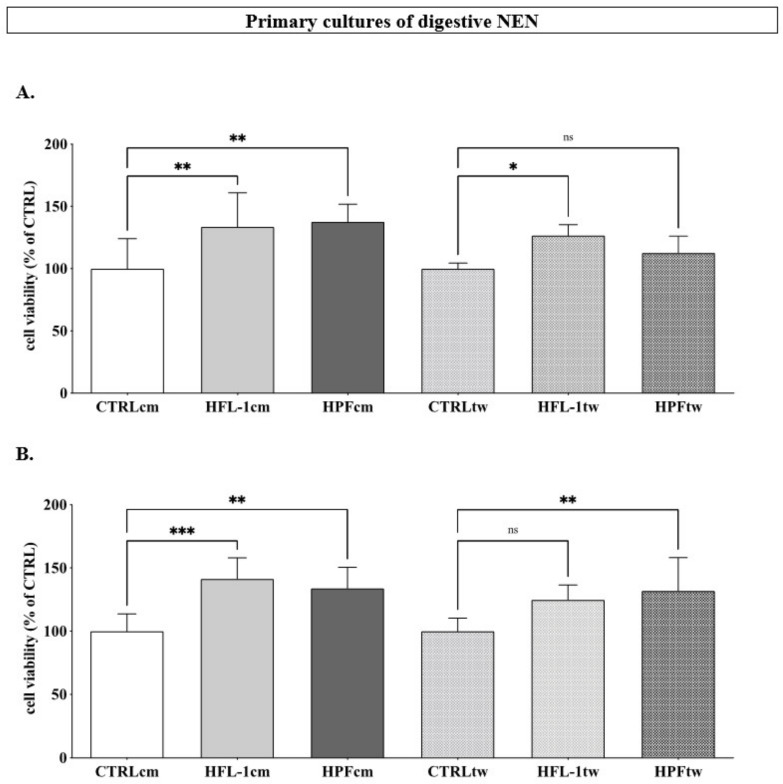
Impact of fibroblasts (conditioned medium (HFL-1cm/HPFcm) and transwell (HFL-1tw/HPFtw)) on the cellular viability of primary culture of a human ileal (**A**) and pancreatic (**B**) neuroendocrine neoplasms. *** *p* < 0.005; ** *p* < 0.01; * *p* < 0.05; ns: not significant.

**Figure 6 cancers-14-03481-f006:**
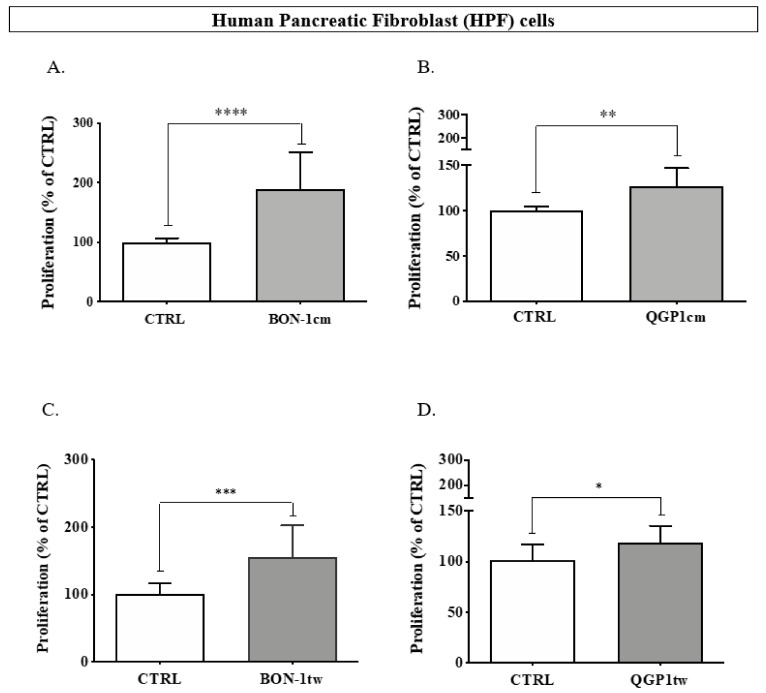
Impact of BON-1cm (**A**) and QGP-1cm (**B**) on the proliferation of HPF cells. Similar experiments were conducted with BON-1 (**C**) and QGP-1 (**D**) in transwells. **** *p* < 0.0001; *** *p* < 0.001; ** *p* < 0.01; * *p* < 0.05.

**Figure 7 cancers-14-03481-f007:**
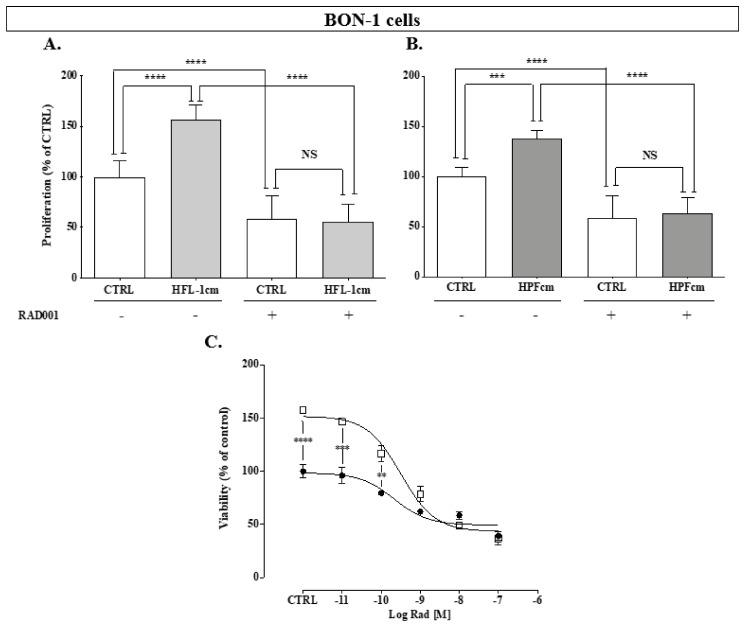
Effect of everolimus (RAD001) on the proliferation of human neuroendocrine cell lines. BON-1 cells were either not treated (CTRL) or treated with RAD001 at 1 nM in the presence of HFL-1cm (**A**) or HPFcm (**B**). Dose-efficacy curve for RAD001 over BON-1 incubated either with (empty squares) or without (solid circles) HPFcm (**C**). Effect of the drugs on the proliferation of QGP-1 incubated either with HFL-1cm (**D**) or HPFcm (**E**). Similarly, a dose-efficacy curve for RAD001 over QGP-1 incubated either with (empty squares) or without (solid circles) HPFcm is shown in panel (**F**). **** *p* < 0.0001; *** *p* < 0.001; ** *p* < 0.01; * *p* < 0.05; NS: not significant.

**Figure 8 cancers-14-03481-f008:**
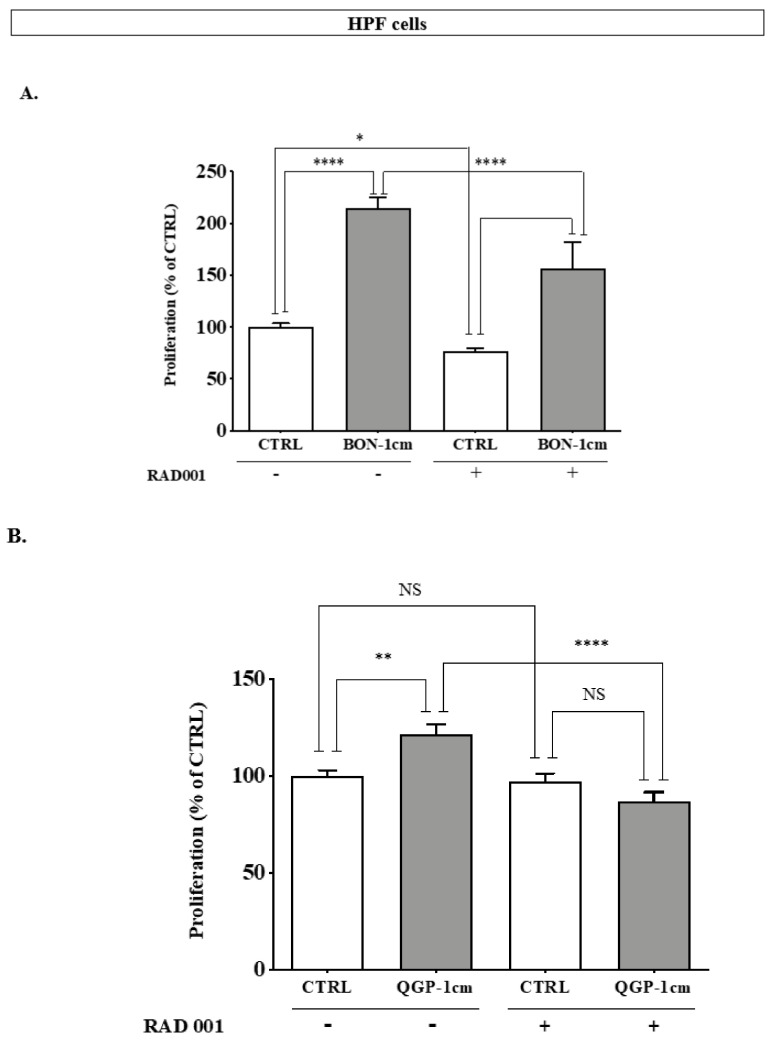
Effect of RAD001 (10 nM) on the proliferation of HPF cells in the presence or absence of BON-1-conditioned medium (**A**) or QGP-1-conditioned medium (**B**). **** *p* < 0.001; ** *p* < 0.01; * *p* < 0.05.

**Figure 9 cancers-14-03481-f009:**
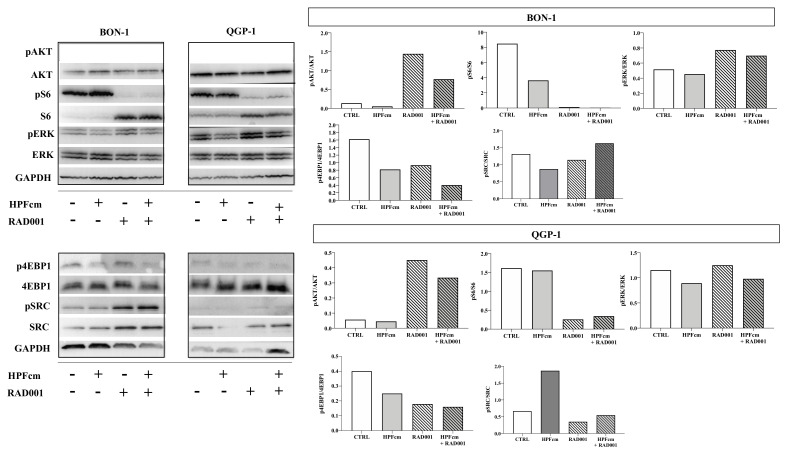
A representative Western blot showing phospho-AKT on Ser473 (pAKT) and total AKT, phospho-ERK1/2 on Thr202/Tyr204 (pERK) and total ERK1/2, phospho-S6 on Ser235/236 (pS6) and S6, phospho-4E-BP1 (p4EBP1) on Ser65 and total 4E-BP1, phospo-SRC on Tyr416 (pSRC) and total SRC and GAPDH performed in BON-1 and QGP-1 cells after incubation with HPFcm and with or without RAD001 at 1 nM and 5 nM, respectively, in comparison to untreated cells. Incubation with RAD001 decreased phosphorylation of S6 in both cell lines.

**Figure 10 cancers-14-03481-f010:**
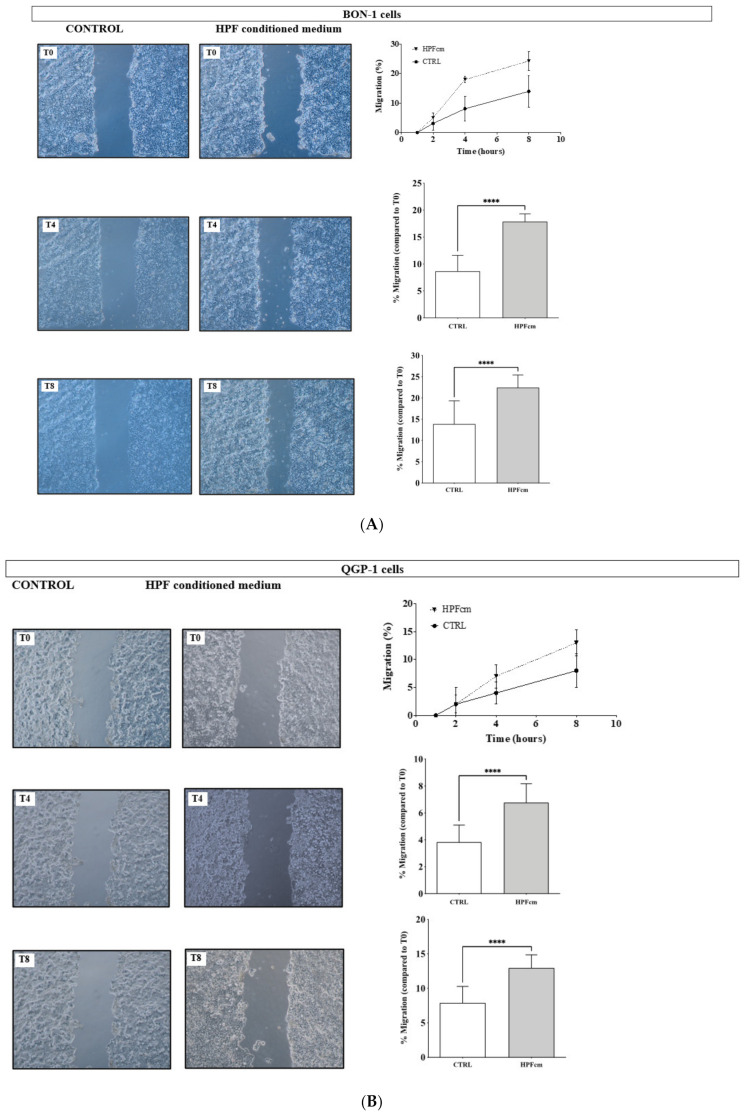
Impact of HPF-conditioned medium on the migration potency of BON-1 (**A**) and QGP-1 cells (**B**). (**A**) Pictures taken at T0, T4 and T8 for BON-1 during the migration assay showing the scratch made by the pipet point. Cells were incubated either with DMEM/F12 + 0.2% BSA (control) or with DMEM/F12 + 0.2% BSA + HPFcm in a 1:1 ratio (HPFcm). Note the migrating cells between the two banks. The migration assay during the time of the experiment (upper right panel) and migration assay at T4 (middle panel) and T8 (lower panel), respectively, compared to T0 in control and HPFcm conditions. (**B**) Pictures taken at T0, T4 and T8 for QGP-1 following the same protocol. Migration assay of QGP-1 during the time of the experiment (upper right panel) and migration assay at T4 (middle panel) and T8 (lower panel), respectively, compared to T0 in control and HPFcm conditions. **** *p* < 0.0001.

## Data Availability

Not applicable.

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
