# Peer review of "Reciprocal Interactions between Fibroblast and Pancreatic Neuroendocrine Tumor Cells: Putative Impact of the Tumor Microenvironment"

_cancers, 2022, doi:10.3390/cancers14143481_

Round 1

Reviewer 1 Report

The authors have significantly improved the manuscript.

Reviewer 2 Report

The study by Cuny et al. is interesting and well written. Characterization of TME interactions is crucial to understanding tumour biology and planning innovative treatments. The Authors build a model based on the investigation of interactions between human pancreatic neuroendocrine neoplasms (PNEN) cell lines and human fibroblast cell lines. Functional effects are described including proliferation, colony formation, and migration. The stimulatory effects mediated by fibroblast in favor of PNEN, although to different extents, were correctly evidenced and mitigated by everolimus. This revised version of the manuscript has been methodologically improved by the addition of the number of experiments and the description of controls. Relevant is the new description of the effect of fibroblasts and everolimus in modifying the signaling pathways (Pi3K/AKT/mTOR and ERK) in PNEN cell lines. The experiments are logically presented and clear. The Discussion is more detailed, critical, balanced, and consistent with the presented data. I believe that the manuscript is interesting and adds suggestions, hypotheses, and stimuli for further research to the scientific community. 

This manuscript is a resubmission of an earlier submission. The following is a list of the peer review reports and author responses from that submission.

Round 1

Reviewer 1 Report

In the present study Cuny and colleagues investigated the reciprocal interaction between human PNEN cells and fibroblasts, in order to explore whether factors secreted by the matrix cells may support the tumour cells growth and invasion and modify the sensitivity to the mTOR inhibitor Everolimus.

Overall, the study aim is clear, original and experiments are soundly performed with clear figures.

The main findings are that proliferation and invasion of pNEN cells (CLS and primary cultures) are increased by cocolture with fibroblasts CLSand viceversa. Everolimus is able to counteract this stimulatory effect.

While I believe that the paper is of potential interest, in my opinion it lacks any attempt of  investigating the underlying mechanisms in terms of involved signaling. In this view some of the authors conclusions seem overemphatized. 

Also, as Everolimus seem able to reverse the stimulatory effect of fibroblasts, I do not agree with the conclusion of the authors that "Fibroblasts, in the TME of PNEN, represent a target of interest to control escape from mTOR inhibitors". Indeed, the authors themselves demonstrate that "Neither the incubation with HFL-1cm or HPFcm resulted in a rescue of the BON-1 cells from the inhibitory effect of RAD001. 

I do have some major and minor changes to suggest before the paper may be considered for publication.

  1. The authors should perform some attempt at least to investigate how the effect of cocolture stimulates the growth of NEN cells. They may perform WB for read-outs of the mTOR pathway (pS6 and 4ebp1) on non stimulated and stimulated cells, and also estimate possible resistance mechanisms (pAKT). Other potentially interesting mechanisms of the cross talk may have to do with the capacity of fibroblasts to activate Src (PMID: 33123993). The Src pathway has been already associated with pNEN and resistance to mTOR inhibitors. Similarly, overactivation of MYC has been associated to resistance to Everolimus (PMID: 32615570) and this may be easily investigated and discussed.
  2. The BON and QGP cells tend to respond in a different manner to Everolimus; QGP are almost resistant. Indeed the authors employ different concentrations in the two cell lines. A dose-efficacy curve with different doses for both CLs in the presence/absence of cocolture would clarify and this needs discussion. Please employ nM to highlight concentration. 
  3. BON cells are extremly sensitive to mTOR inhibitors but tend to lose this characteristic upon chronic exposure (REF 25 in the paper, PMID: 25026292, PMID: 32615570). The authors should test whether a chronic exposure to Everolimus affects the reported interactions.
  4. Add a colony formation assay at least for BON1, with/without previous cocolture and with/without Everolimus.

An in depth discussion on organoids as possible future step to investigate the present issue would be desirable.

Reviewer 2 Report

The major drawback in the manuscript is the conclusions made by the authors and the data shown do not completely correlate. Conclusions were derived based on data that is not very prominent as they have explained in the text. At least few different experiments could have been done to support their hypothesis. I do not recommend the manuscript for publication in Cancers journal. Following are the major issues to be addressed.

  1. In Fig 2.E and 2.F the images of BON-1 cells clearly show that the cell growth prominently increased in the presence of HPF. However, the column graphs in Fig. 2A-D do not show a robust change in the proliferation. Part of the reason is the assay they have selected to demonstrate this phenomenon. I would recommend showing the proliferation rate of the cells as a growth curve over a particular period of time. The author could simply count cells every day and compare the growth rate. Or I would suggest determining the doubling time of the BON-1 cells in the presence of fibroblasts.

  1. Same issue in Fig. 3. I would suggest doing a different assay to match with the cell images.

  1. In Fig. 4 CTG is not a valid assay to show cell proliferation, because it doesn’t give a dynamic range. Again I would suggest growth rate or cell cycle or similar analysis.

  1. In Fig.5B and 5D I do not see a prominent change in the proliferation of HPF cells in the presence of QGP-1. But this was never addressed in the text.

  1. In Fig.6, the data shows that the increase in proliferation of BON-1 and QGP-1 cells in the presence of fibroblasts is mitigated by RAD001. But, the conclusion in the abstract is misleading regarding the use of RAD001.

  1. It would be helpful to label the name of the cell lines on each graph. I could see that in few figures but not all.  

  1. In the methods section, please check the IC50 value of RAD001 on QGP-1 cells. It is mentioned 5X10^9 mol/L.

Reviewer 3 Report

The article is well written and presented. The model is simple and intuitive. However, I recommend to lower the tones of the conclusions since no mechanistic data are reported. In particular, no molecular assessments of mTOR-dependent downstream pathways have been done. The study is predominantly descriptive and hypothesis-generating. ln each section of the Methods I recommend to declare how many times experiments have been done. Positive controls in functional experiments are missing.